# XA4C: eXplainable representation learning via Autoencoders revealing Critical genes

Qing Li[1], Yang Yu[2], Pathum Kossinna[1], Theodore Lun[1], Wenyuan Liao[2]*, Qingrun Zhang [1,2,3,4]*

**1** Department of Biochemistry & Molecular Biology, University of Calgary, Calgary, Canada, **2** Department of Mathematics and Statistics, University of Calgary, Calgary, Canada, **3** Alberta Children's Hospital Research Institute, University of Calgary, Calgary, Canada, **4** Arnie Charbonneau Cancer Institute, University of Calgary, Calgary, Canada

* wliao@ucalgary.ca (WL); qingrun.zhang@ucalgary.ca (QZ)

**Data Availability Statement:** XA4C is publicly available in our GitHub: https://github.com/QingrunZhangLab/XA4C TCGA: https://portal.gdc.cancer.gov/, BRCA: https://portal.gdc.cancer.gov/

## Abstract

Machine Learning models have been frequently used in transcriptome analyses. Particularly, Representation Learning (RL), e.g., autoencoders, are effective in learning critical representations in noisy data. However, learned representations, e.g., the "latent variables" in an autoencoder, are difficult to interpret, not to mention prioritizing essential genes for functional follow-up. In contrast, in traditional analyses, one may identify important genes such as Differentially Expressed (DiffEx), Differentially Co-Expressed (DiffCoEx), and Hub genes. Intuitively, the complex gene-gene interactions may be beyond the capture of marginal effects (DiffEx) or correlations (DiffCoEx and Hub), indicating the need of powerful RL models. However, the lack of interpretability and individual target genes is an obstacle for RL's broad use in practice. To facilitate interpretable analysis and gene-identification using RL, we propose "Critical genes", defined as genes that contribute highly to learned representations (e.g., latent variables in an autoencoder). As a proof-of-concept, supported by eXplainable Artificial Intelligence (XAI), we implemented eXplainable Autoencoder for Critical genes (XA4C) that quantifies each gene's contribution to latent variables, based on which Critical genes are prioritized. Applying XA4C to gene expression data in six cancers showed that Critical genes capture essential pathways underlying cancers. Remarkably, *Critical genes has little overlap with Hub or DiffEx genes*, *however*, *has a higher enrichment in a comprehensive disease gene database (DisGeNET) and a cancer-specific database (COSMIC)*, *evidencing its potential to disclose massive unknown biology*. As an example, we discovered five Critical genes sitting in the center of Lysine degradation (hsa00310) pathway, displaying distinct interaction patterns in tumor and normal tissues. In conclusion, XA4C facilitates explainable analysis using RL and Critical genes discovered by explainable RL empowers the study of complex interactions.

## Author summary

We propose a gene expression data analysis tool, XA4C, which builds an eXplainable Autoencoder to reveal Critical genes. XA4C disentangles the black box of the neural

projects/TCGA-BRCA; COAD: https://portal.gdc.cancer.gov/projects/TCGA-COAD; KIRC: https://portal.gdc.cancer.gov/projects/TCGA-KIRC; LUAD: https://portal.gdc.cancer.gov/projects/TCGA-LUAD; PRAD: https://portal.gdc.cancer.gov/projects/TCGA-PRAD; THCA: https://portal.gdc.cancer.gov/projects/TCGA-THCA; DisGeNET: https://www.disgenet.org/search. Concept Unique Identifier (CUI) used for six cancers are listed as following: BRCA: C0678222, C0006142; COAD: C0009402, C0699790; KIRC: C1378703, C0007134, C0740457; LUAD: C0684249; PRAD: C0600139; THCA: C0549473 DESeq2: http://www.bioconductor.org/packages/release/bioc/html/DESeq2.html WGCNA: https://cran.r-project.org/web/packages/WGCNA/index.html COSMIC Cancer Gene Census: https://cancer.sanger.ac.uk/census.

**Funding:** Q.Z. is supported by an NSERC Discovery Grant (RGPIN-2018-05147), a University of Calgary VPR Catalyst grant, and a New Frontiers in Research Fund (NFRFE-2018-00748). W.L. is partly supported by an NSERC CRD Grant (CRDPJ532227-18). Q.L. is partly supported by an Alberta Innovates LevMax-Health Program Bridge Funds (222300769). The computational infrastructure is funded by a Canada Foundation for Innovation JELF grant (36605) and an NSERC RTI grant (RTI-2021-00675). The funders had no role in study design, data collection and analysis, decision to publish, or preparation of the manuscript.

**Competing interests:** The authors have declared that no competing interests exist.

network of an autoencoder by providing each gene's contribution to the latent variables in the autoencoder. Next, a gene's ability to contribute to the latent variables is used to define the importance of this gene, based on which XA4C prioritizes "Critical genes". Notably, we discovered that Critical genes enjoy two properties: (1) Their overlap with traditional differentially expressed genes and hub genes are poor, suggesting that they indeed brought novel insights into transcriptome data that cannot be captured by traditional analysis. (2) The enrichment of Critical genes in a comprehensive disease gene database (DisGeNET) and cancer-specific database (COSMIC) are higher than differentially expressed or hub genes, evidencing their strong relevance to disease pathology. Therefore, we conclude that XA4C can reveal an additional landscape of gene expression data.

## Introduction

Machine learning (ML) models play increasingly important roles in gene expression analyses. Among many ML techniques, representation learning (RL) has the potential to bring a breakthrough, due to its ability to deconvolute nonlinear structures and denoise confounders [1]. For example, in the data preprocessing stage, in contrast to traditional statistical models that remove principal components to adjust data [2], modern tools can run RL to eliminate noise and possibly non-linear structures in the data [3–7]. Autoencoders (AE) have been extensively utilized to develop various tools for processing expression data [3–5, 8]. A notable feature of AEs is their ability to learn the hidden representations of input data despite the input being noisy and heterogeneous, leading to "latent variables" that are cleaner and more orthogonal for next stages of analysis.

However, such learned representations, although enjoying desirable statistical properties, are difficult to interpret. Therefore, in practice, researchers and clinical practitioners are left with manual inspection of data to decide whether to conduct experimental follow-up or clinical investigations. Additionally, traditional expression analyses naturally provide a list of prioritized genes. For instances, one may identify genes of importance such as Differentially Expressed genes (DiffEx) based on individual genes' marginal effects [9–11] and Differentially Co-Expressed genes [12] based on genes' pairwise correlations [13–15]. Also, Hub genes can be prioritized based on the connectivity of the nodes in a gene-gene co-expression network [13]. In contrast, the (usually uninterpretable) learned representations do not offer analogues for the experimentalists or clinicians to follow up. Although several tools supporting interpretable ML models are available [4, 8, 16], they do not offer individual candidate genes based on learned representations. In particular, Hanczar and colleagues employed gradient methods to analyze the contribution of specific neurons in the network [16], which, however, does not focus on the contribution of individual input genes. Dwivedi and colleagues analyzed the effect of an input feature by differentiating the outcome by switching off the focal input feature, i.e., gene [4]. Although aiming to relieve the problem of interpretation, this method only focuses on the marginal effect of each gene, which does not employ the latest development in eXplainable Artificial Intelligence (XAI) that systematically examines complex models. Recently, Withnell et al proposed XOmiVAE [8] that also employs SHAP and AE, however it focuses more on the classification of samples, instead of prioritizing individual genes for further analysis. Other efforts using XAI in the field of cancer and health outcomes [17–19] also do not prioritize individual genes. Therefore, to facilitate broader use of RL of expression data, interpretable tools that prioritize candidate genes are urgently needed.

Herein, supported by state-of-the-art development in XAI [20], we developed a tool, XA4C (eXplainable Autoencoder for Critical genes), to facilitate explainable analysis and prioritization of individual genes. Technically, XA4C is composed of two main components: First XA4C offers optimized autoencoders to process gene expressions at two levels: whole transcriptome (global) autoencoder, and single pathway (local) autoencoders (**Materials and Methods**). Second, using SHapley Additive exPlanations (SHAP) [21–23], a pioneering method inspired by the popular economic concept of "Shapley Value" quantifying the contribution of a player in a game, XA4C quantifies individual gene's contribution to the learned latent variables in an autoencoder (**Materials and Methods**), and aggregate them to form "Critical index" for each gene (**Materials and Methods**). These Critical indexes will be used to prioritize Critical genes based on user specified cutoff, e.g., 1%.

The term "*Critical gene*", reflecting genes substantially explaining of latent variables is comparable to the popular term "Hub gene" which is defined as the genes with high connectivity in a co-expression network [13]. These two could be considered in parallel because "Critical genes" and "Hub genes" both play a sensible role in gene-gene interactions, although in different forms of representations in an interaction network. In other words, "Hub genes" contribute to the surrounding genes in the co-expression network through correlations in the plain representation, whereas "Critical genes" contribute to linked genes through the latent variables in an autoencoder through explaining their variations. As hidden states presumably incorporate complex correlation structures, the links between genes and hidden states may be considered the analogue of links in a co-expression network, hence the analogous relationship between Hub genes and Critical genes.

By applying XA4C to cancer data offered by The Cancer Genome Atlas [24], or TCGA (**Materials and Methods**), we revealed sensible genes and pathways through SHAP-based explanations. We also carried out thorough investigations of Critical Genes by generating summary statistics in comparison to other conventional means, i.e., DiffEx genes and Hub genes. We observed important properties of Critical genes: first, the overlaps between Critical genes with DiffEx or Hub genes are quite low; and second, Critical genes' enrichment in a comprehensive disease-gene database is higher than the ones of DiffEx and Hub genes. These indicate that Critical genes indeed revealed new candidates into the pathology, and they are even more sensible than the DiffEx and Hub genes revealed by traditional analysis. As a step further, we analyzed the data and discovered Critical genes (that are not DiffEx or Hub) altering the interaction patterns in Lysine degradation (hsa00310) pathway.

## Results

### The XA4C model

Autoencoder (AE) is a RL model that learns representations of input datasets in an unsupervised way [1, 25]. Based on an AE with fully connected neural network, XA4C first learns representations of gene expressions (**Materials and Methods**; **Fig 1A**). Briefly, input gene expression profiles are passed through the encoder network to learn low-dimensional representations through a bottleneck. The decoder is symmetrical to the encoder counterpart to recover the gene expressions. The loss function of training the parameters in the AE is Mean Squared Error (MSE) between input and output, the default for continuous variables. To quantify each gene's contribution to the latent variables, XA4C employs eXtreme Gradient Boosting (XGBoost) [26], an ensemble tree model [27] between the input genes and latent variables (**Materials and Methods**; **Fig 1B**). Then, Tree SHAP explanation [23] is used to assess the contribution of inputs to representations (**Materials and Methods**; **Fig 1B**). As such, XA4C outputs SHAP values for input gene expression individually and quantifies the contribution of

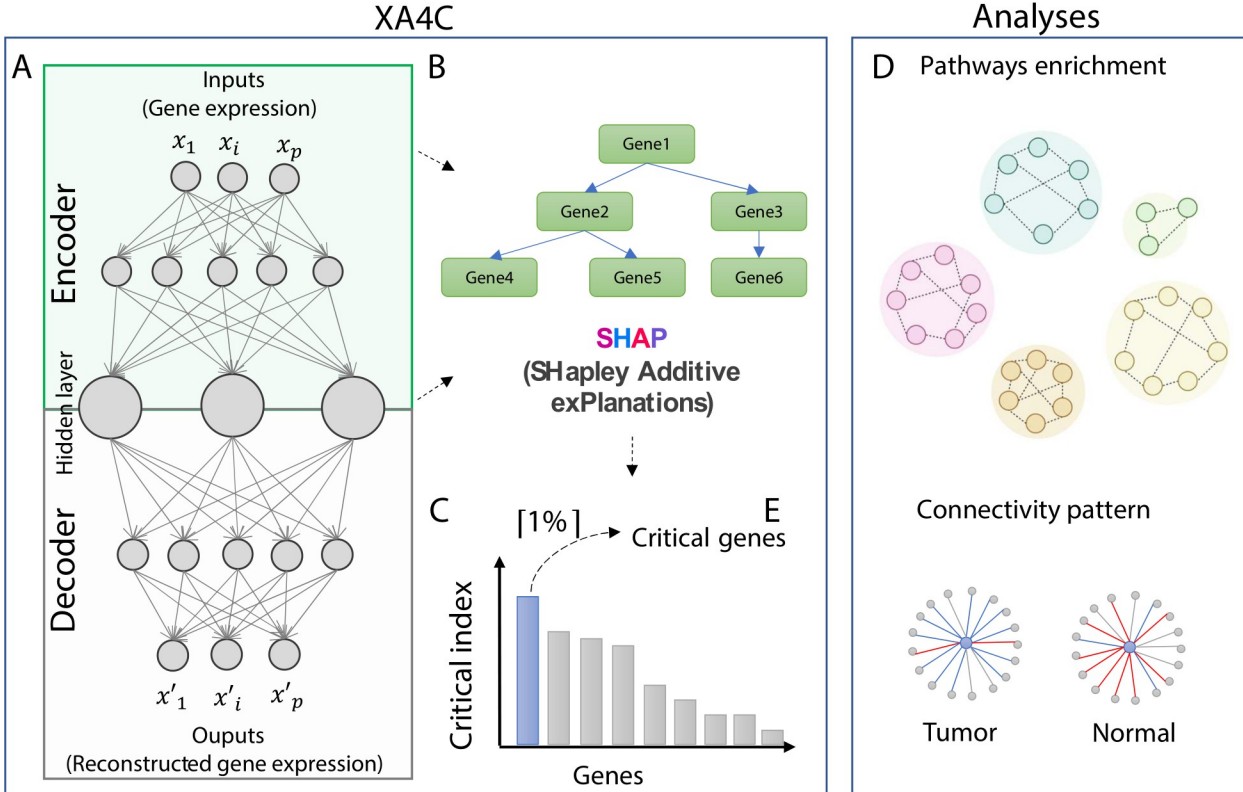

**Fig 1. The XA4C model and potential downstream analysis.** (A) An autoencoder is constructed to learn representations (i.e., latent variables) of input gene expression profiles. (B) XGBoost and TreeSHAP are utilized to evaluate SHAP values and Critical indexes for all genes. (C) Critical genes are the ones with the top 1% Critical indexes. (D) KEGG pathway enrichment identifies sensible pathways overrepresented by prioritized genes with SHAP values. (E) Connectivity analysis discloses interaction patterns among genes centered by Critical genes in pathways.

each input to each representation (i.e., latent variable). Using these SHAP values, XA4C further quantifies the Critical index of a gene by averaging the absolute values of its SHAP value to all latent variable via all the samples (**Materials and Methods**). By ordering these genes based on their Critical indexes, XA4C achieves a list of *Critical genes* above a user-specified cutoff, e.g., top 1% (**Materials and Methods**; **Fig 1C**). As downstream applications of XA4C, the genes together with their Critical indexes may be used for pathway enrichment analysis and connectivity analysis (**Materials and Methods**; **Fig 1D and 1E**).

## Whole-transcriptome AEs with high compression ratio reconstructed with high accuracy

First, we established whole transcriptome AEs based on curated 15,000 genes (**Materials and Methods**). Second, we evaluated reconstruction performances for AEs by calculating $R^2$ from testing samples (**Table 1**). It is observed that AEs with 32 latent variables conducted decent reconstruction with $R^2$ values from the testing dataset varying from 0.42–0.69, which are comparable to AEs with 512 latent variables ($R^2$ values range from 0.50–0.72). However, the compression ratio of AEs with 32 latent variables ($468 \approx 15,000/32$) is 16 times of AEs with 512 nodes ($29 \approx 15,000/512$), which indicates latent variables from the former compress way more information. Therefore, for the whole transcriptome analysis of XA4C, we used AEs with 32 latent variables nodes for the six cancers. We also set up this as the default in XA4C.

**Table 1. Test R² for whole transcriptome autoencoders.** L is the number of layers and H is the number of latent variables.

| Cancer | Primary Tissue | Sample size | Number of genes in AE inputs | Test R² (L = 5, H = 32) | Test R² (L = 1, H = 512) |
|---|---|---|---|---|---|
| BRCA | Breast | 1048 | 15,488 | 0.50 | 0.50 |
| COAD | Colon Adenocarcinoma | 389 | 14,211 | 0.47 | 0.50 |
| KIRC | Kidney | 481 | 14,459 | 0.64 | 0.62 |
| LUAD | Bronchus and lung | 485 | 15,341 | 0.42 | 0.53 |
| PRAD | Prostate gland | 459 | 14,843 | 0.52 | 0.53 |
| THCA | Thyroid gland | 451 | 14,500 | 0.69 | 0.72 |

## Whole-transcriptome Critical genes and pan-cancer pathways

We calculated Critical indexes for the ~15,000 genes (S1 Table) and illustrated the top 30 in Fig 2A. It is observed that the maximum absolute values for Critical indexes range from 0.03 to 0.06, and they decrease quickly. The overall distribution of the Critical indexes all genes and Critical genes in six cancers are in Fig 2B.

We further conducted pathway over-representation analysis on genes with non-zero Critical indexes to reveal sensible pathways underlying cancers (Materials and Methods). We found a handful of pathways that mediate crucial roles in multiple cancers (Fig 3A and S2 Table). Notably, oxidative phosphorylation (OXPHOS) was enriched for five cancers (BRCA, COAD, KIRC, LUAD and THCA), and growing evidence indicate it was an active metabolic pathway in many cancers [28]. Higher expression of OXPHOS genes predicts improved survival in some cancers [29], but also confers chemotherapy resistance in others [30, 31]. Many recent studies proposed to treat this pathway as an emerging target for cancer therapy [29–31]. Another interesting pathway, glutathione metabolism, has been found in two cancers (KIRC and THCA). Glutathione (GSH) is an important antioxidant that maintains cellular redox homeostasis and detoxifies carcinogens [32, 33]. However, GSH metabolism can also play a pathogenic role in cancer by conferring therapeutic resistance and promoting tumor progression [33, 34]. There are novel therapies that target the GSH antioxidant system in tumors to increase treatment response and decrease drug resistance [33]. Furthermore, amino sugar and nucleotide sugar metabolism pathway has been identified LUAD, and studies showed that arresting pathways of carbohydrate metabolism such as central carbon metabolism in cancer, aerobic glycolysis, and amino sugar and nucleotide sugar metabolism can introduce apoptosis in breast cancer [35]. Other well-known cancer pathways, such as apoptosis, p53 signalling pathways, have also been discovered by Critical index-directed enrichment analysis.

As comparisons, we also performed differential expression (DiffEx) analysis [36] and differential co-expression (DiffCoEx) analysis [12] (Materials and Methods). Overall, 31%~83% of XA4C pathways are shared with DiffEx, and the percentage increased to 76%~91% for DiffCoEx (Fig 3B). Our results showed that XA4C's ability of identifying pathways has a larger overlap with the network-based approach DiffCoEx than the marginal effect-based approach DiffEx. It may because AE can handle nonlinear relationships among features, whereas traditional DiffEx analysis couldn't utilize gene network information.

## Overview of within-pathway Critical genes

To further explore the Critical genes within known pathways, we applied AEs to expressions of genes within individual pathways. From the KEGG [37], we downloaded 334 pathways whose numbers of genes varies from dozens to hundreds (S3 Table). For each pathway of each cancer, we constructed an AE, with a small number of latent variables (H = 8) and few layers

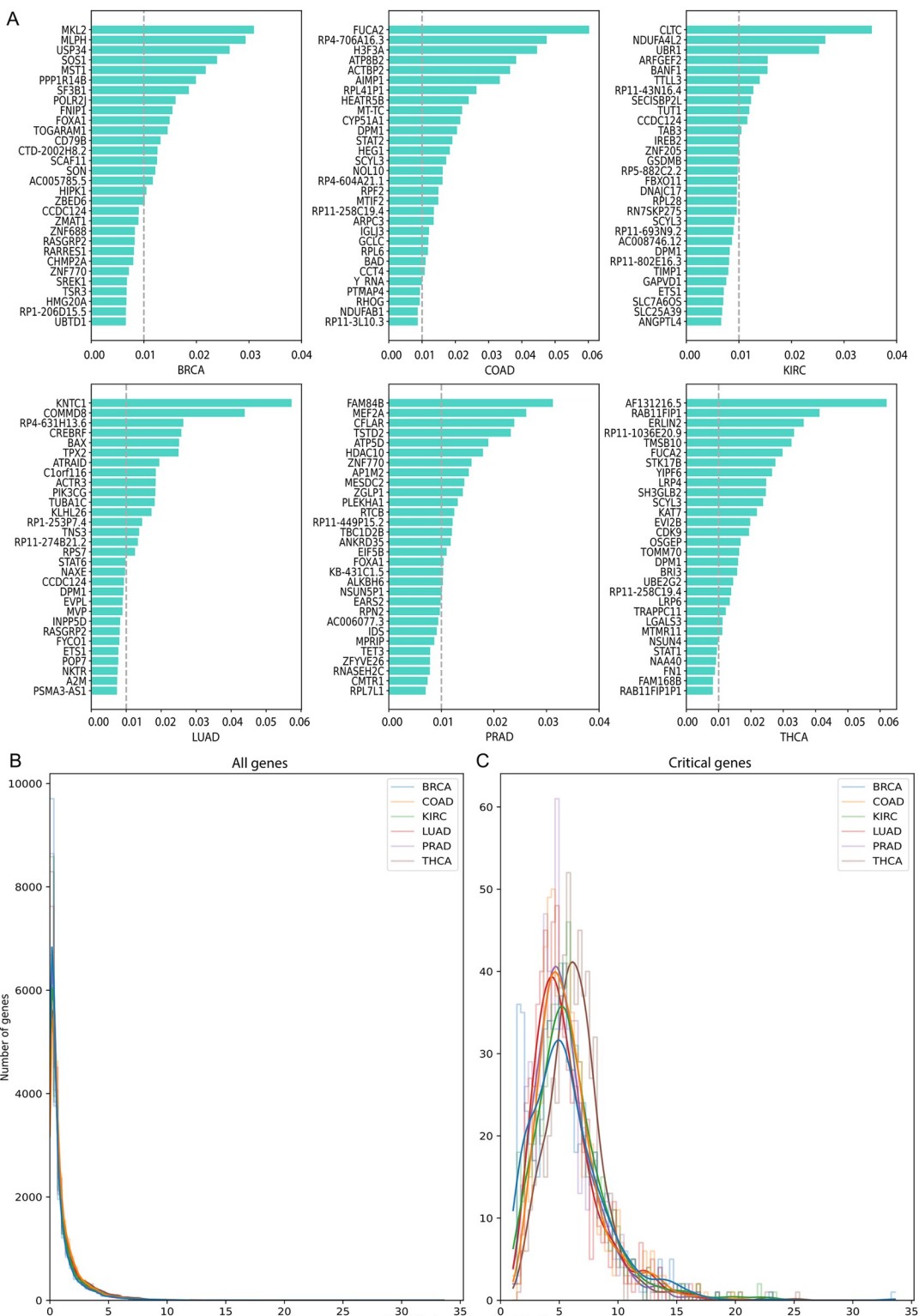

**Fig 2. Whole transcriptome Critical indexes of genes in six cancers.** (A) Genes with the largest 30 Critical indexes summarized among all latent variables and averaged across samples. (B) Distribution of whole transcriptome Critical indexes for all genes. (C) Distribution of whole transcriptome Critical indexes for Critical genes.

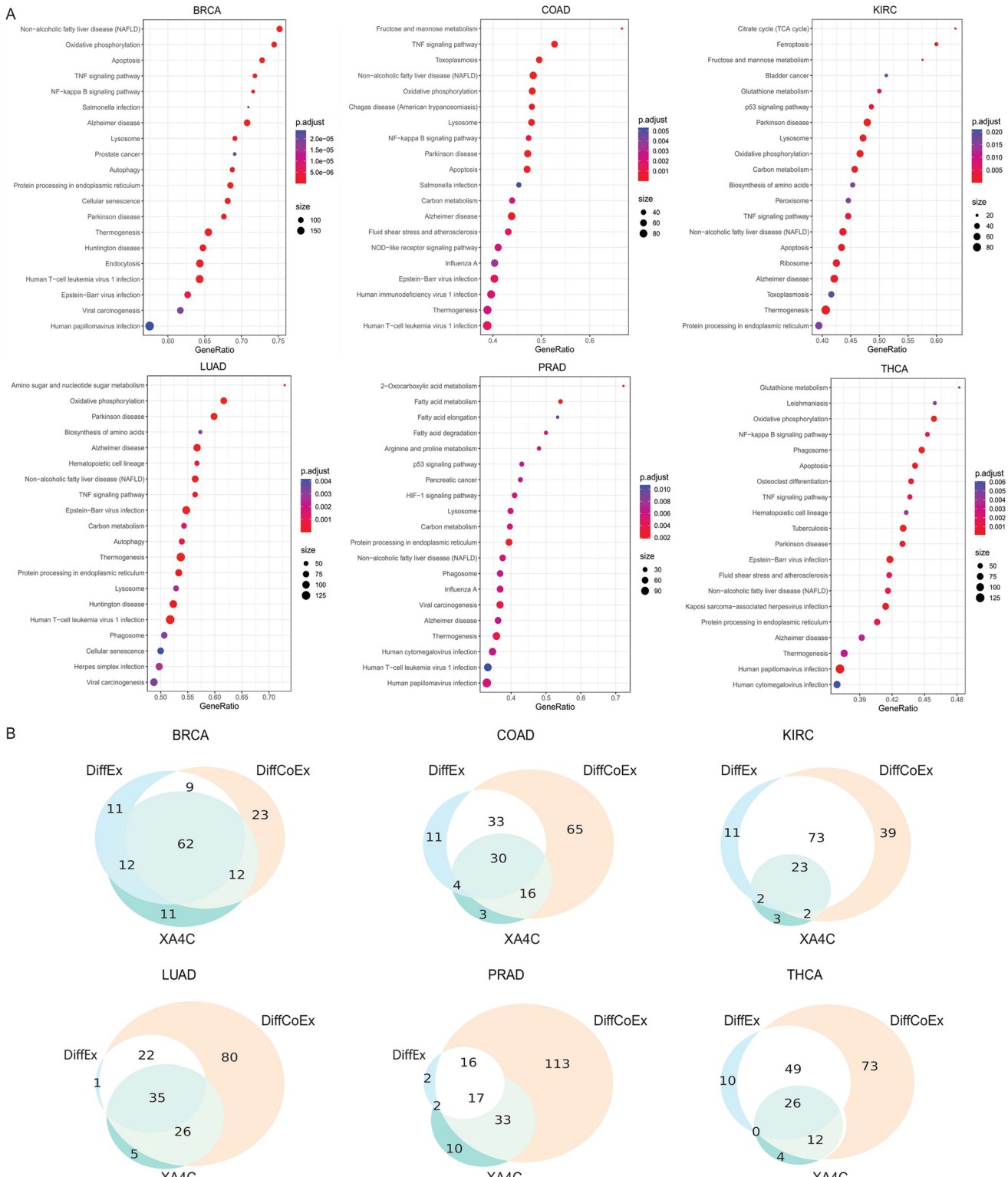

**Fig 3. Pathway enrichment of whole-transcriptome genes.** (A) Top 20 KEGG pathways enriched by genes with non-zero Critical indexes. The p-values are listed in S2 Table. (B) Comparison of pathways enrichment of genes prioritized by XA4C, DiffEx analysis and DiffCoEx analysis.

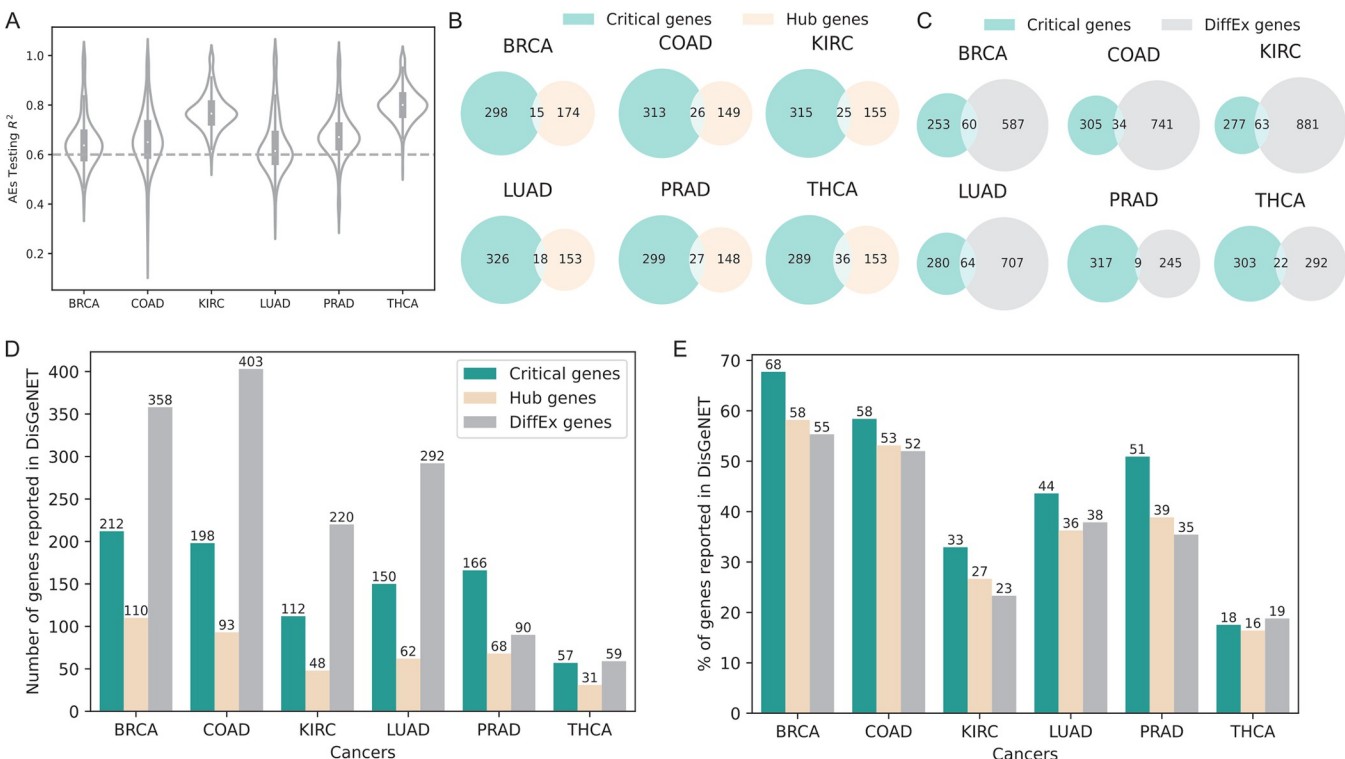

**Fig 4. Generation and analysis of within-pathway Critical genes.** (A) Distribution of $R^2$ (in testing samples) of pathway AEs in six cancers. (B) Overlaps between Critical genes and Hub genes (identified by WGCNA). (C) Overlaps between Critical genes and DiffEx genes. (D) Numbers of Critical, Hub, and DiffEx genes validated by DisGeNET. (E) Percentage of Critical, Hub, and DiffEx genes validated by DisGeNET.

(number of layers L = 3 or 2). We set L = 2 when the number of genes in a pathway is less than 100, or L = 3 otherwise. Pathway AEs testing $R^2$ values have noteworthily high values with mean values above 0.6 for all six cancers (**Fig 4A**). It is also noted that the mean testing $R^2$ values from KIRC, PRAD, and THCA pathway AEs are larger than BRCA, COAD and LUAD, which is consistent with the AEs performance of whole-transcriptome analysis. The distributions of Critical indexes for genes from 334 pathways are similar to whole-transcriptome results (**S1A Fig**) and pathway Critical genes (the largest 1%) mean values also approximate transcriptome Critical genes (**S1B Fig**).

## Critical genes are highly enriched in cancer-related mutations

We conducted a comprehensive analysis of genetic and epigenetic mutations in the critical genes identified by XA4C. We first obtained information from the COSMIC database [38]: the genetic mutations, including missense mutations and copy number variations, and epigenetic mutations, including differential methylation. Based on the available mutation information, we observed a significant proportion of Critical genes (70% averaged for six cancers) that exhibited gained or lost copy number variations. Additionally, approximately 25% of the critical genes showed differential methylation, characterized by a beta-value difference larger than 0.5 compared to the average beta-value across the normal population. Furthermore, around 12% of the Critical genes displayed missense mutations, which have the potential to alter the function of the encoded proteins. The detailed results are listed in **S4 Table.**

## The overlaps between Critical genes and Hub or DiffEx genes are poor

To reveal whether Critical genes indeed make differences in practice, we compare Critical genes to Hub genes defined by Weighted Correlation Network Analysis (WGCNA) [13] as well as DiffEx genes generated by DESeq2 [36] (**Materials and Methods**). As WGCNA outputs 1 Hub gene per pathway and our 1% Critical index cut-off in the pathway is on average approximately 1 or 2 genes, this analysis yields comparable number of genes. We found that, in all six cancers, the overlap between Critical genes and Hub genes are poor (**Fig 4B**) Similar observation is also evident for DiffEx genes (**Fig 4C**). These results show that Critical genes indeed provide a different angle for researchers to analyze expression data.

## Critical genes have higher enrichment in disease genes than Hub and DiffEx

Having learned the low overlap presented above, we then continue to learn whether the Critical genes prioritized by XA4C are indeed sensible. We then examined the DisGeNET [39], a comprehensive database for the enrichment of these three categories of genes (**Materials and Methods**). We noticed that Critical genes are highly enriched in genes with susceptibility reported in DisGeNET. Although DiffEx has the number of successfully validated genes due to its overall substantially more input candidates (**Fig 4D**), Critical gene is the winner when comparing the ratio (numbers of validated genes divided by numbers of input genes) (**Fig 4E**). Notably, the high proportions are consistent across all six cancers, indicating that Critical genes are fundamental for all cancers. We also tried to replicate such enrichment analysis in the COSMIC database that focuses more on mutations. The success rates of all three methods (Critical, Hub and DiffEx genes) are low (**S5 Table**). However, the limited data still indicates the advantage of Critical genes which has higher success rate in majority of cancers compared to alternatives (**S5 Table**).

Using DisGeNET as gold-standard, we also quantitatively calculated the confusion matrices as well as precision, recall, F1-score and accuracy for all three methods (**Materials and Methods**). The results, presented in (**S6 and S7 Tables**), demonstrate that XA4C outperforms the other methods in terms of F1-score (largely contributed by its supremacy on precision), and the accuracy of the three methods are comparable.

Together, these results indicate that XA4C-derived Critical genes capture additional information other than marginally altered gene expression and genes with a high degree of connectivity in the protein-protein interaction network.

## Critical genes alter interaction patterns in a pan-cancer pathway

To further investigate the functions performed by Critical genes in their pathways, we calculated the Pearson correlation between the Critical genes and all the other genes in corresponding pathways. We found that Critical genes display distinct interaction patterns in the co-expression network between tumor and matched normal tissues. We specifically picked up an example in Lysine degradation (I00310) pathway as the Critical genes in this pathway are neither Hub nor DiffEx genes in five cancers. Evidently, the interaction patterns of Critical genes and surrounding genes are dramatically different in five cancers, which are visible by inspections and quantified by the Kolmogorov-Smirnov test (**Fig 5**). First, the intensity of correlation is notably weaker in tumor while compared to normal (lighter color in tumor and brighter color in normal). Moreover, the variance of correlations (reflected by the height of the boxplots) is substantially larger in normal tissues in all cancers except for THCA. Overall, the dramatically changed interaction patterns suggest that Critical genes involved in disease

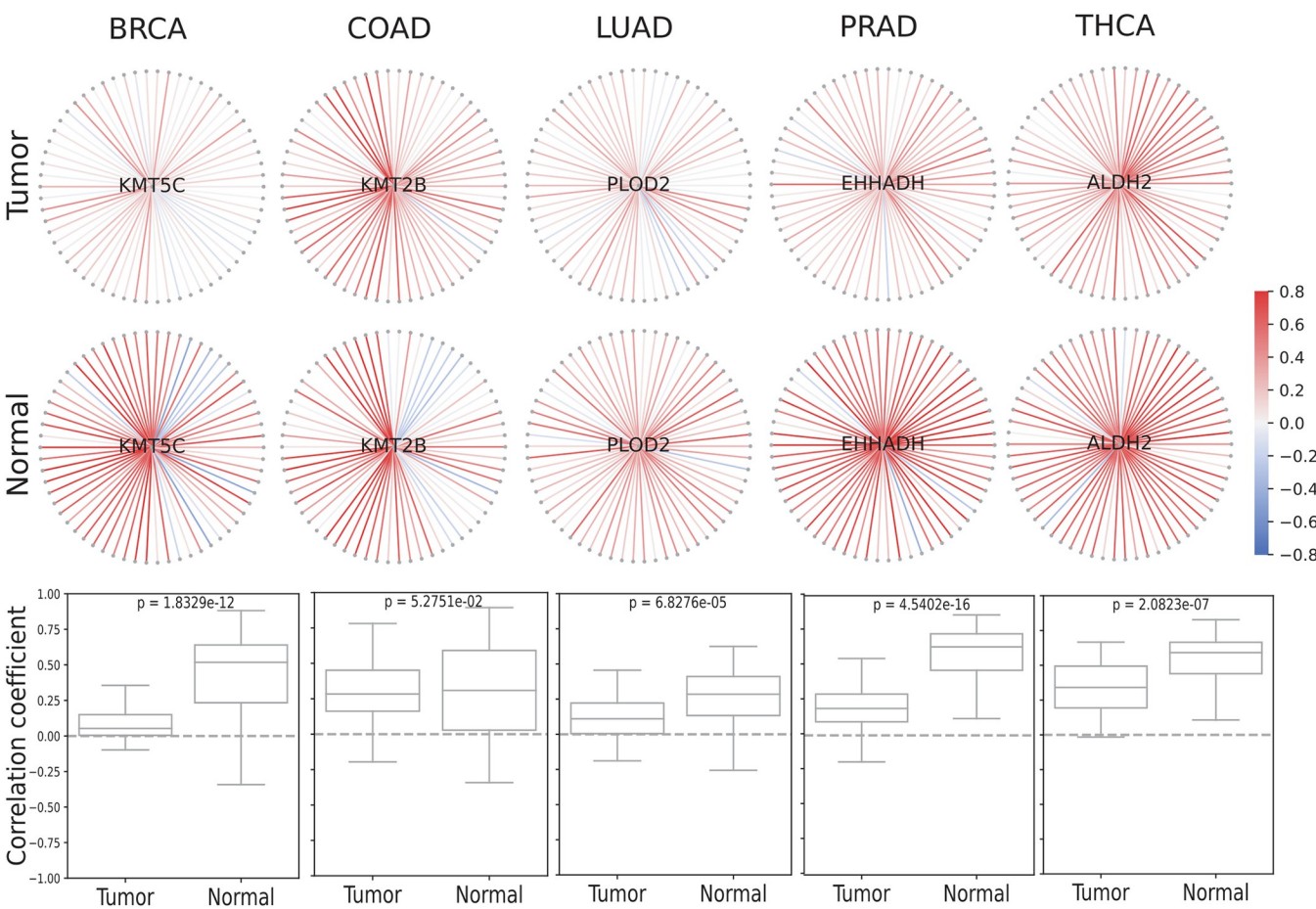

**Fig 5. Critical genes show distinct co-expression networks in tumor and normal tissues.** The Lysine degradation pathway (I00310) is used. Critical genes (light blue) are located at the core of the network, surrounded by additional genes from the same pathway (gray). The boundaries of Pearson's correlation coefficients range from +0.8 (red) to -0.8 (blue). Boxplots show the distributions of two sets of correlations (tumor vs. normal) together with the P-value of the Kolmogorov-Smirnov test, with the null hypothesis being that the two samples were chosen from the same distribution. Critical genes shown in this figure are novel as they have not been identified by traditional analysis search for Hub nor DiffEx genes.

pathogenesis through interactions with other genes although themselves are not marginally differentially expressed (DiffEx genes) nor most connected with other genes (Hub genes) in the pathway.

## Discussion

In this work, we proposed XA4C, an XAI empowered AE tool to support explainable representation learning (RL). A notable contribution of XA4C is the definition of Critical genes, which formed the RL analogue to Hub/DiffEx genes, bringing an additional perspective in characterizing transcription data. By applying XA4C to cancer transcriptomes, XA4C generated Critical indexes and the list of Critical genes. Analyses show that Critical genes are quite different to DiffEx and Hub genes offered standard analysis based on plain representations, opening a potential landscape for in-depth analysis of complex interactions using RL. Impressively, Critical genes enjoy higher success rate in DisGeNET, a comprehensive disease gene database, indicating that Critical genes indeed play functional roles in diseases. As an example of interactions revealed by Critical genes, XA4C highlighted interesting Critical genes playing central roles in pathways by showing distinct the interaction patterns between tumor and normal tissues.

Machine learning algorithms may run into overfitting. In XA4C, there are two models used: Autoencoder and TreeSHAP. The autoencoder by itself is unsupervised, therefore, it may not run into overfitting [40, 41]. More importantly, a sparsity penalty with L1 regularization is applied to XA4C autoencoder loss function, which penalizes non-zero activations. This sparsity penalty can prevent overfitting to some extent because it makes the autoencoder prefer to activate only a subset of its nodes. It also helps generalization by preventing the model from remembering noisy or irrelevant patterns in the training data [42, 43]. It is important to note that TreeSHAP itself does not introduce overfitting if the underlying tree model is not overfitting. In our study, we employed the XGBoost regression model as the tree model. XGBoost models also incorporate regularization techniques to prevent overfitting [44, 45]. With the regularization penalty in both the autoencoder and TreeSHAP, we believe the overfitting is under control in our XA4C model.

To compare whether our protocol using Autoencoder + SHAP indeed outperforms the built-in feature importance values generated by other models, we conducted some comparisons to two representative tools, namely Random Forest and XGBoost. We trained classifiers using Random Forest and XGBoost using established tools [26, 46] and then use their corresponding feature importance values to define "Important genes" analogue to our procedure of defining Critical genes using SHAP values (**Materials and Methods**). We observed that, despite the classification accuracy of Random Forest or XGBoost being excellent (**S8 Table**), the "Importance genes" prioritized by Random Forest or XGBoost (listed in **S9 Table**) have much less functional enrichment in both DisGeNET and COSMIC databases (**S10 Table**).

We acknowledge limitations of XA4C, which could be addressed by our future work. First, we utilized fixed architectures of AEs in this study. Although we compared the performances of AEs on several different architectures and selected the optimal architectures for whole-transcriptome (L = 5, H = 32) and pathway AEs (L = 3 or 2, H = 8), we only tested a few combinations of the architecture parameters. A sophisticated way is to incorporate Bayesian Hyperparameter Optimization[47] for extensively evaluating the combinations of L and H. Second, only conventional AEs are used. It is straightforward to extend to other AEs, such as Variational AE for causal inference and Graph AE to learn causal structure, which is our planned future work. Third, TreeSHAP was utilized in this study to explain the tree model built on AEs inputs and representations. Some SHAP explainers, such as DeepSHAP [48], might be more efficient for neural network models since they omit the phase of constructing Tree models between inputs and representations. Fourth, we summarized SHAP values over samples to generate one Critical index to rank the importance of genes. This gives us general information but ignores the heterogeneity between individual samples. Our future study will focus on individuals to uncover individualized Critical genes to support precision medicine. Finally, XA4C currently supports only transcriptome data, and a natural extension will be the incorporation of additional omics data, such as proteomics and metabolomics.

## Materials and methods

### XA4C model (I): Autoencoder (AE) architectures and parameters tuning

AE consists of two main components: an encoder and a decoder. The encoder converts the input data into a lower-dimensional representation, known as the "latent variables". The decoder then reconstructs the original input data based on latent variables. The objective of an autoencoder is to minimize the reconstruction error between the input data and the reconstructed. From a mathematical perspective, the encoder part can be represented by the encoding function h = f(x), and the decoder can be represented by the decoding function r = g(h). Thus, the whole AE can be described by a function $r = g(f(x))$, where the output r is

reconstructed as approximate as possible to the original input x. XA4C utilizes Mean Square Error $MSE = \frac{1}{n} \sum_{i=1}^{n} (x - r)^2$ as the loss function.

By default, XA4C specifies an AE with 5 coding layers and 32 latent variables for whole transcriptome, and Aes with 3 coding layers (or 2 if the number of genes is lower than 100) and 8 latent variables for pathway gene expressions. We achieved these optimal parameters by testing configurations striking a balance between fewer AE parameters (layers, nodes) and a larger reconstruction $R^2$ in testing sample. To train AE models, we partitioned tumor samples into training and testing datasets with the ratio of 8:2. Genes with median expression levels larger than 1.0 were retained as input. Raw gene expressions were transformed using Log (base 2) function, and then further rescaled to a range between 0 and 1 to match the sigmoid activation function in AE neural networks. To train AE models, we utilized the *Adam* optimizer [49] with a learning rate = $1.0 \times 10^{-3}$ and decay = 0.8 for 500 epochs. The training stops when testing $R^2$ does not increase by more than 0.02% in 10 epochs, or the maximum number of training epochs (3,000) is reached. The batch size was set as the input sample size. These AE models were implemented using PyTorch [50].

## XA4C model (II): Shapley Additive exPlanation (SHAP) framework, XGBoost and TreeSHAP

SHAP is a classical post-hoc explanatory framework to calculate the contribution of each input variable in each sample learn the explanatory effect [21]. Let *f* be the original model and *f(x)* the predicted values. *g(x′)* is the explanation model used to match the original model *f(x)*. Note that explanation models often use simplified inputs *x′* that map to the original inputs through a mapping function $x = h_x(x′)$. XA4C incorporates XGBoost Regressor [26] to represent the model *f* and Tree Explainer [23] as the model *g* to interpret *f* in SHAP. We choose XGBoost because models built through gradient boosting algorithm gives more importance to functional features and are less vulnerable to hyperparameters initializations [51]. Each latent variable has an XGBoost model. TreeSHAP is a SHAP method designed for the black-box tree models including Random Forest, XGBoost, and CatBoost, etc [52]. For each XGBoost regression model (established between all inputs and a latent variable in the AE), XA4C carries out TreeSHAP calculation, leading to a SHAP value for each triple of [gene, latent variable, sample].

## XA4C model (III): definition of Critical indexes and Critical genes

The Critical index of a gene is the weighted average of its SHAP value contributed to all samples and all latent variables. To calculate the Critical index ($WSV_i$) of gene *i*, XA4C first takes the mean absolute value for the gene *i* over all *n* samples, leading to the SHAP value of a gene to a latent variable:

$$SHAP \ of \ gene \ i \ for \ latent \ variable \ h: \ SV_{h,i} = \frac{1}{n} \sum_{s=1}^{n} |SHAP \ value_{h,i,s}|$$

Then, XA4C summarizes SHAP values ($SV_{h,i}$) across all latent variables (e.g., 32 in the default AE configuration of XA4C) and weights them by $w_h$, which is the XGBoost regressor $R^2$ of the *h*-th latent variable:

$$Critical \ index \ of \ gene \ i: \ WSV_i = \sum_{h=1}^{H} w_h \ SV_{h,i}$$

XA4C ranks all genes based on their Critical indexes and take the top ones (based on a user-specified cutoff) as Critical genes. By default, XA4C selects 1% for both whole transcriptome is and within-pathway analysis. The ceiling function $\lceil * \rceil$, i.e., the least integer greater than the actual value (e.g., $\lceil 0.2 \rceil = 1, \lceil 1.8 \rceil = 2$), is used when the cut-off is not an integer.

## TCGA gene expression data for six cancers

We utilized gene expression (number of genes M = 56,497)) from six cancers (BRCA: breast invasive carcinoma, COAD: colon adenocarcinoma, KIRC: kidney renal clear cell carcinoma, LUAD: lung adenocarcinoma, PRAD: prostate adenocarcinoma, THCA: thyroid carcinoma) from The Cancer Genome Atlas (TCGA) [24]. The raw count data was downloaded from the TCGA data portal and then converted to Transcripts Per Million (TPM) gene expression matrices using gene lengths obtained through the BioMart package (code available in our GitHub). We then performed basic quality control to examine the overall structure of the data by PCA analysis [53] and did not find unexpected data points. In particular, we curated genes with median expression levels higher than 1.0, which reduces interference from lowly expressed genes and the number of genes decreased to around 15K. We also performed log-2 transformation to these 15K genes. When applying XA4C to cancer data, for each cancer type, one whole-transcriptome AE and 334 pathway AEs were trained independently. The architecture of the embedding network contains 32 latent variables for whole-transcriptome AE or 8 latent variables for pathway AEs. These whole-transcriptome or pathway representations and their corresponding inputs were passed through the XGBoost regression model on which TreeSHAP SHAP values for inputs genes.

## Pathway over-representation analysis

Over-representation analysis (ORA) [54] is a statistical method to understand which biological pathways may be over-represented. It determines whether genes from pre-defined sets (e.g., those belonging to a specific KEGG pathway) are present more than would be expected (over-represented). The p-value can be calculated by the hypergeometric distribution:

$$p = 1 - \sum_{i=0}^{k-1} \frac{\binom{M}{i}\binom{N-M}{n-i}}{\binom{N}{n}}, \text{ where } N \text{ is the total number of genes in the background set,}$$

$n$ represents the size of the list of genes of interest (for example the number of DiffEx or Critical genes), $M$ stands for the number of genes annotated background, and $k$ is the number of genes in the list annotated to the gene set. The background distribution, by default, is all the genes that have annotation, which are KEGG pathway genes in this study. We utilized the ORA analysis provide by the R package named WebGestalt [55].

## Differential expression genes, Differential Co-expression genes and hub genes Analysis

Differential expression (DiffEx) analysis aims at identifying differentially expressed genes between experimental groups. We utilized DESeq2 [36], which tests for differential expression by negative binomial generalized linear models [36], with default parameters. DESeq2 employs a generalized linear model framework with a negative binomial distribution to assess differential expression between two groups. Initially, it estimates the fold change for each gene between the groups, and subsequently calculates the Wald test statistics and corresponding p-values. These p-values reflect the level of evidence contradicting the null hypothesis that there is no disparity in gene expression between the conditions. These p-values are further adjusted for

multi-test correction, and the significance level (alpha) utilized is 0.05, a conventional parameter for statistical tests.

As DiffEx analysis is based on linear models, it ignores the non-linearity displayed by gene expression. Therefore, we used differential co-expression networks to identify groups (or "modules") of differentially co-expressed genes utilizing "DiffCoEx" [12], an extension of the WGCNA[13]. DiffCoEx begins with the construction of two adjacency matrices: $C_{case} : c_{ij}^{case} = corr(gene_i, gene_j)$ for case samples and $C_{control}$ similarly for control samples. DiffCoEx used the Spearman rank correlation, and a matrix of adjacency difference is then calculated:

$$D : d_{ij} = \sqrt{\frac{1}{2}|sign\left(c_{ij}^{case}\right)*(c_{ij}^{case})^2 - sign\left(c_{ij}^{control}\right)*(c_{ij}^{control})^2|}^{\beta}$$

where $\beta \geq 0$ is an integer tuning parameter which can be selected in multiple ways. In this study, we chose $\beta \in [5,6,7,8,9,10]$ such that we achieved minimum number of modules with the largest module containing the smallest number of genes. Next, a Topological Overlap dissimilarity Matrix is calculated where smaller values of $t_{ij}$ indicates that a pair of genes $gene_i$, and $gene_j$ have significant correlation changes (between case and control).

$$T : t_{ij} = 1 - \left(\frac{\sum_k d_{ik}d_{kj} + d_{ij}}{\min(\sum_k d_{ik}, \sum_k d_{kj}) + 1 - d_{ij}}\right)$$

Finally, the dissimilarity matrix $T$ is to identify DiffCoEx genes.

Hub genes, highly connected within biological networks, were identified using gene co-expression networks constructed with WGCNA. We utilized the "chooseTopHubInEachModule" function from the WGCNA R package, applying it to the gene expression matrix from pathways. Default settings were maintained, with the power parameter of 2 and the type parameter of "signed."

## Performance measurements of accuracy and sensitivity

In this evaluation, we compare the performance of Critical genes, Hub genes, and DiffEx genes using quantified performance measurements. We first construct confusion matrices. Considering DisGeNET-reported genes as the gold-standard. For a particular tool (critical genes, hub genes, or DiffEx genes), we defined true positives (TP) as genes identified by the tool and are reported by DisGeNET, true negatives (TN) as genes not identified and not reported in DisGeNET, false positives (FP) as genes identified but not reported in DisGeNET, and false negatives (FN) as genes identified but not reported in DisGeNET. Based on the confusion matrices, we calculated precision, recall, F1 score and accuracy using the formula below:

$$\text{Precision} = \frac{TP}{TP + FP}$$

$$\text{Recall} = \frac{TP}{TP + FN}$$

$$\text{F1 score} = \frac{preciison \times recall}{precision + recall} = \frac{TP}{TP + \frac{1}{2}(FP + FN)}$$

$$\text{Accuracy} = \frac{TP + TN}{TP + TN + FP + FN}$$

## Comparison to the feature importance values generated by Random Forest and XGBoost

The Random Forest and XGBoost classifiers were trained on gene expressions from 335 pathways. The classifiers were trained using default parameter settings with 500 estimators (number of trees in the forest). To ensure balanced representation of tumor and normal samples, we randomly sampled from tumor samples with the same number of normal tissue samples to construct datasets (S8 Table). The resulting dataset was then divided into training and test datasets in a 7:3 ratio. We performed model optimization on training datasets and evaluated its performance on test datasets. Notably, the classifiers exhibited impressive performance, as indicated by high weighted-averaged F1 scores across the 335 pathways. Specifically, the Random Forest classifier achieved an F1 score of 94% for six cancers, while the XGBoost classifier achieved an F1 score of 92% for the same six cancers (S8 Table). Subsequently, feature importance values were derived from these well-trained classifiers. The same procedure in the identification of Critical genes was used to define the "Important genes", defined as genes with top 1% ceiling importance values from classifiers in each pathway.

## Supporting information

**S1 Fig. Pathway Critical indexes of genes in six cancers.** (A) Distribution of pathway Critical indexes for all genes in the corresponding pathways. (B) Distribution of pathway Critical indexes for Critical genes in the corresponding pathways.
(TIF)

**S1 Table. Critical indexes for whole transcriptome AEs in six cancers.**
(XLSX)

**S2 Table. KEGG pathways enrichment for XA4C, DiffEx and DiffCoEx.**
(XLSX)

**S3 Table. . KEGG pathways and numbers of genes in pathways.**
(XLSX)

**S4 Table. XA4C-identified critical genes in pathways associated with copy number variations, differential methylation, and genetic mutations from COSMIC database.**
(XLSX)

**S5 Table. Number and Percentage of COSMIC Cancer-Specific Census Genes Identified by Critical Genes, Hub Genes, and DiffEx Genes.**
(XLSX)

**S6 Table. Confusion Matrix for Critical Genes, Hub Genes, and DiffEx Genes Using DisGeNET as the Gold Standard.**
(XLSX)

**S7 Table. Evaluation of Classification Performance for Critical Genes, Hub Genes, and DiffEx Genes Using DisGeNET as the Gold Standard.**
(XLSX)

**S8 Table. Weighted-Averaged F1 Scores for 335 Pathways with Train and Test Sample Sizes.**
(XLSX)

**S9 Table. Important (top 1% ceiling) genes identified by Random Forests and XGBoost.**
(XLSX)

**S10 Table. Enrichment of critical genes, hub genes, DiffEx genes, important genes from Random Forests and XGBoost in DisGeNET and COSMIC Cancer Census Genes.** (XLSX)

## Author Contributions

**Conceptualization:** Qingrun Zhang.

**Data curation:** Qing Li, Yang Yu, Pathum Kossinna.

**Formal analysis:** Qing Li, Qingrun Zhang.

**Funding acquisition:** Wenyuan Liao, Qingrun Zhang.

**Investigation:** Qing Li, Qingrun Zhang.

**Methodology:** Qing Li, Qingrun Zhang.

**Resources:** Wenyuan Liao, Qingrun Zhang.

**Software:** Qing Li, Pathum Kossinna.

**Supervision:** Wenyuan Liao, Qingrun Zhang.

**Validation:** Qing Li, Qingrun Zhang.

**Visualization:** Qing Li, Theodore Lun.

**Writing – original draft:** Qing Li, Qingrun Zhang.

**Writing – review & editing:** Qing Li, Qingrun Zhang.

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
