## [Decision Letter · Decision Letter 0]

28 Jun 2023

Dear Dr Zhang,

Thank you very much for submitting your manuscript "XA4C: eXplainable representation learning via Autoencoders revealing Critical genes" for consideration at PLOS Computational Biology.

As with all papers reviewed by the journal, your manuscript was reviewed by members of the editorial board and by several independent reviewers. In light of the reviews (below this email), we would like to invite the resubmission of a significantly-revised version that takes into account the reviewers' comments.

We cannot make any decision about publication until we have seen the revised manuscript and your response to the reviewers' comments. Your revised manuscript is also likely to be sent to reviewers for further evaluation.

Sincerely,

Shihua Zhang

Academic Editor

PLOS Computational Biology

Ilya Ioshikhes

Section Editor

PLOS Computational Biology

Reviewer's Responses to Questions

**Comments to the Authors:**

Reviewer #1: Review Gen

This paper addresses the challenge of interpreting learned representations from autoencoders in gene expression analyses. It introduces a novel tool called XA4C (eXplainable Autoencoder for Critical genes), which combines state-of-the-art techniques in eXplainable Artificial Intelligence (XAI) with autoencoders. XA4C employs optimized autoencoders at global and local levels to process gene expressions and utilizes SHapley Additive exPlanations (SHAP) to quantify the contribution of individual genes to the learned latent variables.

Introduction

Overall, the introduction section provides a clear background on the significance of ML models in gene expression analysis, addresses the limitations of existing approaches, and introduces the XA4C tool as a solution for explainable analysis and prioritization of critical genes which is good.

To highlight the novelty of XA4C, it would be helpful to explicitly state how it differs from existing interpretable ML tools and what unique features or capabilities it brings to the field. This will help readers understand the specific contributions of XA4C.

Result

When conducting pathway over-representation analysis, it would be valuable to include statistical significance measures, such as p-values or false discovery rates, to determine the significance of pathway enrichment. This would provide more robust evidence for the involvement of specific pathways in cancer.

General

In order to adhere to the standard practice of writing abbreviations, the author should provide the full name or description of an abbreviation the first time it is mentioned, followed by the abbreviation in parentheses. However, for subsequent mentions of the same abbreviation within the same section or context, it is generally not necessary to repeat the full name or description. Instead, the abbreviation can be used directly.

To address the issue in the provided lines, the author should modify the text as follows:

Line 120 variables. To quantify each gene’s contribution to the latent variables, XA4C employs eXtreme Gradient Boosting (XGBoost)

Line 339: eXtreme Gradient Boosting 19 (XGBoost) Regresso

Line 377: pathway representations and their corresponding inputs were passed through the eXtreme Gradient Boosting (XGBoost)

Reviewer #2: Tha manuscript proposed “Critical genes”, defined as genes that contribute highly to learned representations. Then, applied eXplainable Autoencoder on the genes to find netowrk of genes and highly contributing genes (discriminative genes) for each type of studied cancer (BRCA, etc..). The manuscript is wll-presented, the methods are properly applied. However, I have some minor concerns:

- The literature lack of the recent application of explainableAI in cancer and health outcomes. KI suggest if the authors may highlight studies such as (PMID: 36738712 and/or PMID: 37233630).

-The results does not show any performance measurements such as accuracy, sensitivity, etc.

-AUCROC curve of the predition model may shows the performance of the model. Also how the model avoided over-fitting.

Reviewer #3: XA4C: A Tool for the Identification of Critical Genes in Cancer

Summary

Li et al.'s work is of considerable importance in cancer genomics. Identifying critical genes associated with various types of cancer is crucial for understanding the mechanisms underlying the disease and developing effective therapeutic strategies. The authors have combined autoencoders and the SHAP framework to develop a new computational model, XA4C, which is aimed at extracting hidden features from transcriptome data and determining the contribution of each gene. Integrating these advanced machine learning techniques allows for a more comprehensive analysis and understanding of high-dimensional gene expression data. The ability of XA4C to uncover novel critical genes could potentially contribute to early detection, personalised treatment strategies, and new insights into the biology of cancer.

To enhance the value of this work, the manuscript should incorporate comparisons with existing models, integrates a thorough methodology for gene identification by considering multiple genetic alterations and validate the results with established databases. These additions would enrich the manuscript's quality and fortify its relevance in cancer research.

Comments

1. Consider Multiple Criteria for Identifying Cancer-Related Genes: The manuscript emphasises the use of differential expression in identifying critical genes. However, it is important to acknowledge that differential expression and hub genes are not the only criteria for determining cancer genes. There are various factors, such as changes in DNA methylation, gain-of-function mutations in oncogenes, loss-of-function mutations in tumour suppressor genes, copy number alterations, chromatin accessibility, and changes in protein expression, that also play a role in cancer progression. The authors could enhance the manuscript by analysing whether the critical genes identified through the XA4C model are associated with some or any of these changes in the studied cancer types. This broader approach can provide a more comprehensive understanding of the genes' roles in cancer.

2. Validate Findings with Known Cancer Genes from COSMIC Database: To increase the robustness and credibility of the findings, the authors should consider validating the critical genes identified by the XA4C model against known cancer genes listed in the COSMIC (Catalogue of Somatic Mutations in Cancer) database. By evaluating which among the identified critical genes are classified as Tier 1 and Tier 2 cancer genes in relation to the differentially expressed genes and the hub genes, the authors can provide additional evidence that supports the utility and accuracy of the XA4C model in identifying relevant cancer genes. This validation with a reputable external database would add significant value and trustworthiness to the results presented in the manuscript.

3. Incorporate Comparisons with Existing Models: The manuscript presents the XA4C model, which combines autoencoders and SHAP values to interpret the contributions of individual genes in the context of cancer transcriptome data. It might be beneficial for the authors to include a comparison section where the performance and interpretability of XA4C are rigorously compared to other existing models and techniques in the same field. This will help in validating the robustness and utility of the XA4C model. This could include traditional statistical methods such as XGBoost or Random Forests approaches.

**Have the authors made all data and (if applicable) computational code underlying the findings in their manuscript fully available?**

Reviewer #1: Yes

Reviewer #2: Yes

Reviewer #3: Yes

PLOS authors have the option to publish the peer review history of their article (what does this mean?). If published, this will include your full peer review and any attached files.

Reviewer #1: No

Reviewer #2: No

Reviewer #3: **Yes: **Musalula Sinkala
---

## [Decision Letter · Decision Letter 1]

29 Aug 2023

Dear Dr Zhang,

We are pleased to inform you that your manuscript 'XA4C: eXplainable representation learning via Autoencoders revealing Critical genes' has been provisionally accepted for publication in PLOS Computational Biology.

Best regards,

Shihua Zhang

Academic Editor

PLOS Computational Biology

Ilya Ioshikhes

Section Editor

PLOS Computational Biology

Reviewer's Responses to Questions

**Comments to the Authors:**

Reviewer #2: The authors have addressed the reviewer comments

Reviewer #3: I thank the authors for addressing all my comments and suggestions.

**Have the authors made all data and (if applicable) computational code underlying the findings in their manuscript fully available?**

Reviewer #2: Yes

Reviewer #3: Yes

PLOS authors have the option to publish the peer review history of their article (what does this mean?). If published, this will include your full peer review and any attached files.

Reviewer #2: **Yes: **Abedalrhman Alkhateeb

Reviewer #3: No

---

## [Editor Report · Acceptance letter]

27 Sep 2023

PCOMPBIOL-D-23-00689R1 

XA4C: eXplainable representation learning via Autoencoders revealing Critical genes

Dear Dr Zhang,

I am pleased to inform you that your manuscript has been formally accepted for publication in PLOS Computational Biology. Your manuscript is now with our production department and you will be notified of the publication date in due course.

With kind regards,

Zsuzsanna Gémesi
